# A Transformer-Based Approach to Leakage Detection in Water Distribution Networks

**DOI:** 10.3390/s24196294

**Published:** 2024-09-28

**Authors:** Juan Luo, Chongxiao Wang, Jielong Yang, Xionghu Zhong

**Affiliations:** 1Computer Science and Electronic Engineering, Hunan University, Changsha 410082, China; 2Institute of High Performance Computing, Agency for Science, Technology and Research, Singapore 138632, Singapore; wang_chongxiao@cfar.a-star.edu.sg; 3School of Internet of Things Engineering, Jiangnan University, Wuxi 214000, China; jyang@jiangnan.edu.cn

**Keywords:** water distribution networks, leakage detection, transformer, attention model, convolutional neural network

## Abstract

The efficient detection of leakages in water distribution networks (WDNs) is crucial to ensuring municipal water supply safety and improving urban operations. Traditionally, machine learning methods such as Convolutional Neural Networks (CNNs) and Autoencoders (AEs) have been used for leakage detection. However, these methods heavily rely on local pressure information and often fail to capture long-term dependencies in pressure series. In this paper, we propose a transformer-based model for detecting leakages in WDNs. The transformer incorporates an attention mechanism to learn data distributions and account for correlations between historical pressure data and data from the same time on different days, thereby emphasizing long-term dependencies in pressure series. Additionally, we apply pressure data normalization across each leakage scenario and concatenate position embeddings with pressure data in the transformer model to avoid feature misleading. The performance of the proposed method is evaluated by using detection accuracy and F1-score. The experimental studies conducted on simulated pressure datasets from three different WDNs demonstrate that the transformer-based model significantly outperforms traditional CNN methods.

## 1. Introduction

Water distribution pipes in a city can span hundreds of kilometers and connect thousands of nodes, creating a complex network that facilitates urban water supply. These pipes are usually buried underground, and many have been in service for decades. The aging of pipes and external forces are the main risks, causing significant damage and leakages in WDNs. It is reported that nearly one-third of drinking water is wasted annually due to pipe leakages [1]. Additionally, pipe leakages can lead to drops in in-pipe water pressure, resulting in the intrusion of contaminants and pathogens, which can cause serious public health issues [2]. Detecting these leakages is challenging because leak points are usually not observable and local water loss is difficult to differentiate from highly fluctuating daily demand data. However, advancements in hydraulic sensors and communication technologies have enabled the deployment of comprehensive data acquisition systems. These systems collect large volumes of time-series data, such as pressure and flow, to profile the state of hydraulic dynamics and detect pipe bursts and leakages [3,4], as shown in Figure 1. Solar energy or mains electricity [5] is typically used to power the remote terminal unit to ensure continuous operation.

Based on different sensor types, the current solutions for leakage detection in WDNs can be mainly divided into two categories: (i) flow data-based methods [6] and (ii) pressure data-based methods [7]. For the former one, WDNs in cities are often divided into zones to construct district metering areas (DMAs) [8,9]. Flow meters are employed to record the flow in and out of each zone and generate the total water utilization of a given DMA. The real water consumption is calculated by summing all the readings from consumer meters. By analyzing the difference between these two sets of data, it is possible to determine the volume of non-revenue water, indicating potential leakages in the WDN [10]. However, using DMAs to detect leakage requires an effective network partition scheme [11]. Determining the inlet and outlet pipes is complex in real-world operations, and optimizing DMAs remains a significant challenge. Additionally, synchronizing readings from WDNs and customer meters is difficult, leading to significant errors in non-revenue water estimation. The pressure data-based methods rely on WDN topology and pressure measurements. Leakages are detected either by comparing the measured pressure with predictions computed according to hydraulic models or by estimating pressure drops and identifying corresponding patterns [12].

In the past, various data-driven approaches have been proposed to detect leakages in WDNs. According to data usage, these algorithms can be broadly categorized into signal prediction methods and classification methods. The idea behind prediction methods is to estimate future water demand, consumption, flow, or pressure based on historical data [13]. These predictions are then compared with actual measurements to detect discrepancies, indicating potential anomalies. Typical algorithms employed include the Kalman filter (KF) [14] and its variants [15] and expectation maximization (EM) algorithms [16]. These methods typically assume that the hydraulics are accurately modeled and the system parameters, such as pipe roughness coefficients, are known a priori. In practice, developing a hydraulic model that accurately matches a real WDN system is challenging, and the system parameters can only be roughly estimated and tend to change drastically as pipe age increases. In classification methods, different feature sets are extracted from the collected data, and classifiers are trained to identify the unique characteristics of different event types, such that normal or anomalous events can be categorized for coming measurements [17,18,19]. Recently, popular machine learning methods, such as AEs and CNNs, have been employed for leakage detection in WDNs. In essence, features are extracted by an AE or a CNN, and a loss function is defined to determine whether the coming data represent a leakage pattern. The advantage of classification methods is that they are data-driven and do not require in-depth knowledge about WDN topology and the corresponding parameters. They only need to learn from the historical data regarding the leak and no-leak data patterns [20].

In this paper, we propose a transformer-based leakage detection model that learns the distribution of normal pressure data. Transformers have recently proven to be powerful deep learning models in applications such as speech recognition, natural language processing, and computer vision processing [21,22,23]. Here, we introduce such a model to capture the detailed changes and identify similar patterns in pressure data series. We split the pressure data into sequences, with each sequence representing one week of data (e.g., if one pressure datum is received in half an hour, we have, in total, 336 data a week). The position of each data point within the sequence is encoded as position embeddings, which, together with the pressure data, are fed to the model. An encoder is then applied to learn the contextual features of the pressure data and map them into a latent vector representation. Given the periodic nature of daily water demand and corresponding pipe pressure variations, the transformer’s attention mechanism relates different positions of similar features to compute a representation of the data sequence. The decoder then transforms these latent vectors back into a new sequence by sampling from the latent space. Leakages are identified by calculating the residuals between the generated sequence and the input data.

One advantage of the proposed method is that it can perform classification by using only pressure data, eliminating the need for network partitioning and pipeline parameters. Additionally, it incorporates global information due to the long-term dependencies in the pressure data. Our experimental results show that the proposed method significantly improves leakage detection performance. Figure 2 provides an overview of the architecture of the proposed model. The main contribution of this study is the development of a transformer-based model for detecting leakages in WDNs. We comprehensively study the features of pressure drops and apply data normalization across each leakage scenario to better capture leakage patterns. Furthermore, we concatenate the position embeddings with the pressure data, rather than adding them together within the model, to better capture the correlation between historical and daily interval pressure data.

The structure of the paper is organized as follows: Section 2 introduces related work on leakage detection in WDNs. Section 3 describes the problem of leakage detection, including its challenges and difficulties. Section 4 provides an overview of the proposed model and detailed components. In Section 5, we describe the datasets used in our experiments and the result analysis of this work. Conclusions and some potential future research directions are given in Section 6.

## 2. Related Work

Traditional leakage detection techniques predominantly depend on manual procedures and external auxiliary equipment, such as electromagnetic scanners [24], infrared thermography [25], and ground-penetrating radar [26]. However, employing these techniques for leakage detection typically proves inefficient due to extensive time requirements, labor costs, and limited coverage. Recent advancements in sensor technology and intelligent algorithms have significantly propelled the innovation of leakage detection approaches within WDNs. Recent advancements in leakage detection methods have approached the problem by treating leak and no-leak data as a binary classification issue. Popular machine learning algorithms, such as Principle Component Analysis (PCA) [27], Support Vector Machine (SVM) [28], K-Nearest Neighbors (KNNs) [29], and CNNs [30,31], have been introduced for this purpose. Additionally, Deep Neural Networks (DNNs) have proven to be effective tools for data classification and have been employed for leakage detection [32].

In a study by Leonzio et al. [33], an AE was designed to encode input pressure data into compressed features and then decode the feature back to reconstruct a new data sequence. This approach aims to identify the absence of leaks by ensuring that the output of the trained network closely resembles the input pressure data, thereby flagging any significant deviations as potential leaks. One notable advantage of AEs is their reliance solely on pressure-series data acquired in the absence of leaks. Another study [34] integrated Long Short-Term Memory (LSTM) neural networks into a Recurrent Neural Network (RNN) architecture to discern various patterns in measurement data associated with different types of leakages. In a separate investigation [35], an Artificial Neural Network (ANN) was employed to predict water flow and pressure, and a leakage warning was activated when the variance between the actual and predicted data surpassed a predefined threshold. However, these approaches often depend on local pressure information and fail to learn the long-term dependencies within pressure-series data. To address these limitations, we propose a transformer-based model capable of capturing long-term dependencies within pressure data in our work.

## 3. Problem Definition

In WDNs, leakages can manifest as either incipient or abrupt, as discussed in [36]. Incipient leaks typically develop gradually and persist over time, with relatively small leakage volumes. These leaks may gradually escalate until their detection prompts intervention measures. Conversely, abrupt leaks, such as pipe bursts, release large quantities of water within a short timeframe. Figure 3 shows pressure and demand data in various leakage scenarios. The left column of the figure, comprising Figure 3(a1–d1), represents demand data assumed to remain constant during a leak. In the middle column, Figure 3(a2–d2) depict pressure data without leaks, exhibiting periodic variations corresponding to changes in demand. Figure 3(a3–d3) represent pressure data during leak events. Notably, small leaks may not cause a discernible pressure drop, complicating identification, as seen in Figure 3(a3,d3). Conversely, large leaks typically result in significant pressure drops, facilitating detection, as illustrated in Figure 3(b3,c3).

We receive pressure data every half hour, with each timestamped sequence consisting of *T* data points processed. A sequence of size *T* is structured as
(1)$$\begin{array}{cc} X & =\left[x_{1},x_{2},\ldots ,x_{T}\right], \end{array}$$
where $X\in R^{T\times 1}$, *T* is the length of the sequence, and $x_{t}$, for t=1,2,…,T, denotes the pressure data with index *t* in the pressure series. With a total of 48 data points per day, we employ a window approach for analysis. At time step *t*, we select a window of pressure data containing the last *L* pressure data with a stride of 1. Each analyzed window consists of L−1 historical pressure data and one current pressure reading. The analyzed window at time step *t* is organized as
(2)$$\begin{array}{cc} x_{t} & =\left[x_{t-L},x_{t-L+1},\ldots ,x_{t}\right], \end{array}$$
where $x_{t}\in R^{L\times 1}$, *L* is the window size, $x_{t}$ denotes the current pressure, and $\left[x_{t-L},x_{t-L+1},\ldots ,x_{t-1}\right]$ denotes the historical pressure data. Additionally, $\left[\ldots ,x_{t-48\times 2},x_{t-48},x_{t}\right]$ represents the daily interval pressure data, enabling the comparison of pressure data at the same time across different days. The objective is to train a model capable of distinguishing pressure data patterns during normal operation from those indicative of various leakage scenarios.

## 4. Proposed Method

In this work, we developed a transformer-based model for leakage detection in WDNs, as depicted in Figure 2. The architecture comprises an encoder module responsible for processing the input sequence and extracting latent feature representations, while the decoder module generates a new pressure sequence by dynamically attending to these latent features. To capture the dependency of the pressure series, an attention mechanism is employed, and a sigmoid function serves as the activation function in the transformer. Similar to the approach outlined in [33], we train the model by using no-leak pressure data to reconstruct input pressure without anomalies. Given the varying operating pressures across different WDN scenarios, we normalize the pressure data of each scenario to the range [0,1]. This normalization ensures that all features share the same scale to make the model more robust. Such normalization is defined as
(3)$$x_{t}\leftarrow \frac{x_{t}-\min\left(X\right)}{\max(X)-\min(X)+\epsilon },$$
where min(X) and max(X) are the minimum and maximum values in the time series of each scenario. ϵ is an extremely small constant value to prevent zero division. Many data mining methods learn under the assumption that the training data are independent and identically distributed (i.i.d.) [37]. We use the Kolmogorov–Smirnov (KS) statistics [38] to evaluate whether the sequences are i.i.d. in this work. The two sequences can be assumed to have the same distribution if the probabilistic value (*p*-value) of KS is larger than 0.025 [39]. In Figure 4a,d, the data from the two scenarios are not i.i.d., as the *p*-value equals 0. In Figure 4b,c, the data from the two scenarios are i.i.d., as the *p*-value equals 0.178. This demonstrates that after normalization, the data are i.i.d., which ensures that the datasets have the same distribution before applying the proposed model.

### 4.1. Transformer Model

Each transformer stack in Figure 2 consists of a self-attention network, a normalization layer, a feed-forward network, and key operations. Pressure data serve as the input to the first self-attention network by concatenating the position embeddings. Subsequently, the input and output of the first self-attention network are aggregated via the normalization layer and residual connection, and such a process is repeated for other transformer stacks. We describe the main components of the transformer as follows.

#### 4.1.1. Position Embeddings

In the transformer model, position embeddings play a crucial role in capturing the positional information of trends or patterns within the data. Following the approach outlined in [40], we generate position embeddings by using sine and cosine functions, expressed as
(4)$$\begin{array}{cc} {\mathrm{PE}}_{\left(pos,2k\right)} & =sin(pos/\left(1000^{2k/d_{\mathrm{model}}}\right)), \end{array}$$
(5)$$\begin{array}{cc} {\mathrm{PE}}_{\left(pos,2k+1\right)} & =cos(pos/\left(1000^{2k/d_{\mathrm{model}}}\right)), \end{array}$$
where pos is the data index in a sequence, *k* is the dimension of the input data sequence, and $d_{\mathrm{model}}$ is the dimension of the output embeddings. Each dimension of the positional encoding corresponds to a sinusoid. To preserve the original values of input pressure data, we concatenate the position embeddings with the pressure data $x_{t}$ to form the input to the encoder, defined as
(6)$$x_{t}^{cat}=\begin{array}{c} \left[\begin{array}{c} x_{t}\\ {\mathrm{pe}}_{t} \end{array}\right] \end{array}=\begin{array}{c} \left[\begin{array}{cccc} x_{t-L} & x_{t-L+1} & \cdots & x_{t}\\ {pe}_{t-L} & {pe}_{t-L+1} & \cdots & {pe}_{t} \end{array}\right] \end{array},$$
where $x_{t}^{cat}\in R^{L\times 2}$ is the input to the encoder and ${\mathrm{pe}}_{t}\in R^{L\times 1}$ represents the position embeddings of $x_{t}$. In Figure 5, we compare the original pressure data with data incorporating the position embeddings. The top figure clearly shows a distinct pressure drop indicative of a leakage, easily identifiable. Conversely, the bottom figure displays the input data after combining the pressure data with position embeddings. Despite significant pressure changes, the leakage point appears blurred due to the influence of position embeddings, making detection challenging.

#### 4.1.2. Attention

The attention mechanism serves as the pivotal component within a transformer architecture, enabling each token in a sequence to glean insights from others. This mechanism weighs the importance of different tokens to capture contextual information and long-term dependencies effectively. Mathematically, the attention operation for a *d*-dimensional input feature is defined as
(7)$$\mathrm{Attention}\left(Q,K,V\right)=\mathrm{softmax}\left(\frac{QK^{T}}{\sqrt{d_{k}}}\right)V,$$
where Q, K, and *V* are the query, key, and value matrices, respectively. The operation involves calculating the similarity between *Q* and *K* by using a scaled-dot product, followed by scaling the similarity by $\sqrt{d_{k}}$ to stabilize the weights. The resulting similarity matrix is then multiplied by the value matrix *V* to obtain the weighted attention feature. In the encoder module, the input pressure sequence $X_{t}^{cat}$ is utilized to generate the matrices *Q*, *K*, and *V* for the attention operation. Subsequently, the encoding serves as keys and values for attention operations in the decoder module. Here, the query matrix comprises pressure data with the same timestamps on different days. Figure 6 illustrates the local attention layers, wherein *Q*, *K*, and *V* represent linear transformations of the input sequence. To efficiently handle the input pressure $x_{t}\in R^{336\times 1}$, we convert it into a matrix of size (48×7). Here, 48 denotes the number of pressure data points in a day, while 7 signifies the number of days in a week. The pressure data at corresponding timestamps across different days are utilized as input for the decoder module.

#### 4.1.3. Feed-Forward Network

Let us assume that $x^{\prime }$ is the output of the attention layer. Each stack within our model incorporates a fully connected feed-forward network (FFN), serving as a position-wise function. The FFN comprises two linear transformations, and the first one is followed by a Rectified Linear Unit (ReLU) activation function, defined as
(8)$$y_{t}^{D}=(\mathrm{ReLU}\left(x^{\prime }W_{1}+b_{1}\right))W_{2}+b_{2},$$
where $W_{*}$ and $b_{*}$, with ∗ denoting index 1 or 2, represent the weight and bias parameters, respectively.

To showcase the capability of learning the long-term dependencies in pressure series with the proposed model, we use the t-distributed stochastic neighbor embeddings (t-SNE) tool [41] to visualize the latent features on the Hanio network after the encoder module. As shown in Figure 7, each point represents a latent feature at a time step and the color of points varies according to the change in the time steps. Figure 7a displays the t-SNE embeddings of the latent features with one week of pressure data. It reveals that adjacent latent features are closely situated, indicating a strong correlation between nearby time steps, and vice versa. Figure 7b illustrates the t-SNE embeddings of the latent features at the same time on different days. Subplots A and B depict the embeddings of pressure data at 24:00 and 6:00 separately from Monday to Friday over a two-month period, and subplots C and D give the embeddings of pressure data at 24:00 and 6:00 on the corresponding weekends. It can be observed that the embeddings in each subplot can easily be clustered; hence, the daily periodicity of the data plays an important role in feature learning. In addition, the embedding patterns differ significantly between weekdays and weekends due to variations in water demands. Finally, Figure 7c compares the t-SNE embeddings of pressure data during normal operation and with leakages, denoted by points in green and orange, respectively. The distribution of latent features due to leakage data closely resembles that of normal pressure data, potentially leading to significant loss when attempting to reconstruct pressure from leakage data.

### 4.2. Leakage Detection

After the transformer decoder, we utilize a linear transformation to reconstruct the pressure sequence, given by
(9)$$y_{t}=\mathrm{Sigmoid}\left(y_{t}^{D}W_{3}+b_{3}\right),$$
where the output of the decoder $y_{t}^{D}$ is used as the input of the linear transformation. Subsequently, the mean square error (MSE) loss function is employed to identify the leakages in WDNs. During the training stage, the output signal is generated by using the data from the no-leak scenario, and we compute the peak value of MSE loss for each window.
(10)$$\mathrm{leak}=\left\{\begin{array}{cc} 1, & \mathrm{if}{∥y_{t}-x_{t}∥}_{2}\geq A\\ 0, & \mathrm{otherwise} \end{array}\right.$$
and
(11)$$\begin{array}{c} {∥y_{t}-x_{t}∥}_{2}=\frac{1}{n}\sum_{t=1}^{n}{\left(y_{t}-x_{t}\right)}^{2}, \end{array}$$
where $y_{t}$ and $x_{t}$ indicate the output and input of the decoder module, respectively. We use A as the threshold, which is the argmax value of the loss during training in the no-leak scenario. In Figure 8, we visualize the pressure after normalization and the test loss of one scenario for the proposed model trained on the Hanoi network, showing the pressure and loss for each time step. It is apparent that the loss exhibits a strong correlation with peaks in pressure. Moreover, there is a high correlation of loss across different time steps. This characteristic allows the model to effectively detect leakages in each scenario after being trained on a no-leak scenario.

### 4.3. Evaluation Metrics

In this work, we utilize detection accuracy and F1-score, which are defined below, to evaluate the performance of the proposed model.
(12)$$\mathrm{Accuracy}=\frac{\mathrm{TP}+\mathrm{TN}}{\mathrm{TP}+\mathrm{FP}+\mathrm{TN}+\mathrm{FN}},$$
(13)$$F1=2\times \frac{\mathrm{Precision}\times \mathrm{Recall}}{\mathrm{Precision}+\mathrm{Recall}},$$
and
(14)$$\mathrm{Precision}=\frac{\mathrm{TP}}{\mathrm{TP}+\mathrm{FP}},\mathrm{Recall}=\frac{\mathrm{TP}}{\mathrm{TP}+\mathrm{FN}},$$
where TP (true positive) represents the number of leakages that are correctly detected, and FP (false positive) indicates the number of leakages erroneously assigned, TN (true negative) denotes the number of normal pressure data correctly predicted, and FN (false negative) signifies the number of normal pressure data erroneously assigned. Accuracy measures the proportion of correct detection out of the total leakages, providing an overall assessment of performance. On the other hand, the F1-score considers both false positives and false negatives, providing a balanced evaluation of precision and recall.

## 5. Experiment and Results

### 5.1. Datasets

Three network topologies, Hanoi, Net1, and Anytown, were employed to generate the demand and pressure data in our experiments [42]. An EPANET-compatible Python package named “wntr” [43] was used for simulation and analyzing the WDNs. The pipeline network used in the dataset is provided as an EPANET INP file. The three network topologies are depicted in Figure 9. In these diagrams, the connection points and water sources correspond to junctions in the network. Each junction is equipped with a Pressure Transducer (PT) and a Flow Transducer (FT), which generate pressure and flow data. All junctions are interconnected via pipelines. It is important to note that the model parameters (such as pipe length, diameter, and roughness) vary for each network. Each scenario was generated based on the pressure-driven nodes water demand model (PDD). This approach allowed us to accurately simulate and analyze the behavior of WDNs under different conditions.

At the same time, the periodicity of daily demand was approximated based on the Fourier series of the real historical water demand of Water Supply Enterprise [42]. Figure 10 illustrates the presence of weekly periodic and seasonal trends in the data. The variance attributed to daily usage was modeled by incorporating random noise. The daily periodicity captures the fluctuations in water demand within a week, while the seasonal trend accounts for variations in water demand due to changing seasons. The inclusion of random noise accounts for fluctuations caused by unpredictable factors, such as human activity (e.g., the opening and closing of valves) and pipeline maintenance activities. If a leakage occurs in the pipeline, we observe a drop in pressure first at the junction closest to the leakage point. This pressure transient then propagates throughout the network, reflecting the transient response to the leakage event.

We generated datasets for 10 different scenarios in each network, where one of the datasets contained no-leak data and was used as our training dataset. In the operation of water supply networks, pressure data are typically uploaded to the central station every 30 min or occasionally hourly. For this work, all datasets were generated for 30 min intervals for one year, i.e., 17,520 data points for each dataset. Table 1 provides a detailed description of the leakage scenarios simulated on each network.

### 5.2. Implementations

Based on the aforementioned datasets, we conducted a study to evaluate the performance of the proposed model for leakage detection. We utilized detection accuracy and F1-score (abbreviated as Acc and F1 in the result tables) as performance evaluation metrics. For comparison, we implemented the Autoencoder method described in [33] and the basic self-attention model described in [40]. The AE method has previously been applied to several datasets on the Hanoi network, achieving a detection accuracy of 89%, as reported in [33]. The implementation of the basic self-attention model aims to showcase the advantages of the attention mechanism over the AE method.

The training parameters used in this work include MSE as the loss function and the Adam optimizer with a learning rate of 0.001. We applied an Early Stopping strategy with a patience of 10. Each dataset was trained for 15 epochs with a batch size of 8. All experiments were conducted on a single NVIDIA RTX 3090 GPU.

Regarding our model, we implemented it by adding the position embeddings with pressure data directly, denoted by Propose-1, and by concatenating the position embeddings with pressure data, denoted by Propose-2, to illustrate the performance under different position embedding fusing schemes.

### 5.3. Results

The performance of four methods, namely, AE, Attention, Propose-1, and Propose-2, in different leakage scenarios on three networks (Hanoi, Net1, and Anytown) is shown in Table 2. Overall, the attention-based methods (Attention, Propose-1, and Propose-2) performed better than the AE. Both the accuracy and F1-score of the proposed methods were higher than those of the AE- and basic attention-based methods. Notably, the proposed method that concatenates the position embeddings with pressure data (Propose-2) outperformed the method consisting in simply adding them together (Propose-1).

Table 3 provides the average performance of different methods across the three network topologies. The proposed transformer model, which concatenates the position embeddings with pressure data, achieved the best performance in all metrics on both the Hanoi and Anytown networks. On the Net1 network, the F1-score of the Propose-1 method was slightly higher than that of the Propose-2 method. Overall, the average performance of the proposed methods significantly surpassed that of the AE method [33]. These results underscore the robustness of the proposed methods in detecting both small and large leakages across different network topologies. Furthermore, the proposed methods exhibited better detection accuracy, false alarm rates, and miss detection rates compared with the AE method.

In addition to evaluating different methods, we analyzed the impact of window size on the leak detection results. This analysis was performed by using a random selection of datasets from the Hanoi network. Figure 11a–c give the performance of the implementation using the window size of one day, two days, and one week for various epochs. It is evident that larger window sizes tended to yield better detection performance. The best results were observed when utilizing one week of data. This can be attributed to the fact that richer contextual information can be leveraged when using one week of data, thereby enhancing the capability of the model to detect anomalies. Notably, the performance achieved with a window size of two days was comparable to that of one week when longer epochs were used. In Figure 11d–f, we compare the results from the pressure data collected at the same hour on different days with the window sizes of two days, one week, and one month for various epochs. It can be observed that increased data utilization led to improved detection performance. Specifically, as more data were incorporated, the model became better equipped to discern anomalies within the dataset.

## 6. Conclusions

In this paper, we introduce a transformer-based leakage detection model for WDNs, capitalizing on the strengths of attention architectures to capture long-term dependencies and contemporaneous information in pressure time series. Unlike simply adding the position embeddings and pressure data together, we concatenate them to further enhance detection performance. We evaluated our model on three different WDNs, and the experimental results demonstrate its superiority over the AE method in terms of detection accuracy. Additionally, the F1-score metric was used to validate the model, according to which our model again outperformed the AE method. In our future work, we aim to assess the performance of the proposed method on real data acquired from urban cities. Additionally, we plan to explore the fusion of other types of modality information, such as hydrophone signals, as an avenue for further enhancement.

## Figures and Tables

**Figure 1 sensors-24-06294-f001:**
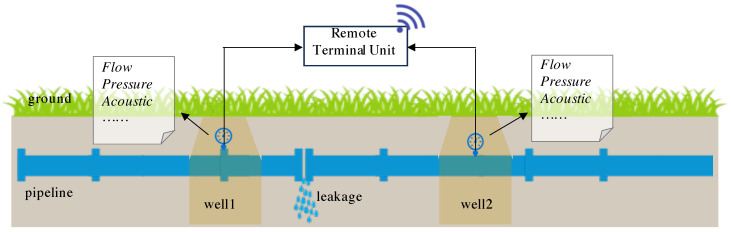
Various sensors are mounted on pipelines to acquire hydraulic data and transmit them to data centers to detect leakages.

**Figure 2 sensors-24-06294-f002:**
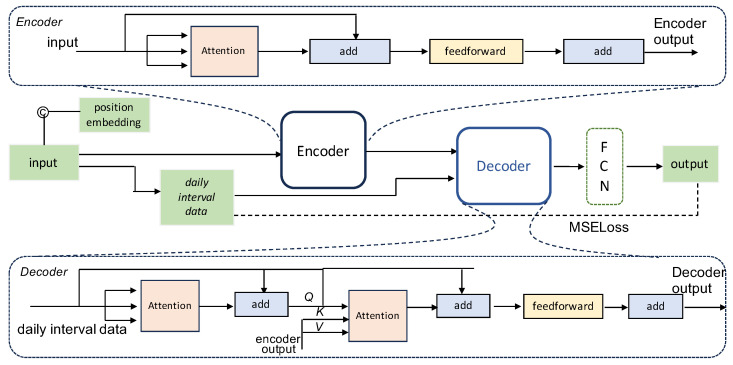
The overall architecture of the proposed model. (1) Instead of performing the direct addition of position embeddings and inputs as in most attention mechanisms, we concatenate position embeddings with pressure data. (2) We consider the dependency of contemporaneous pressure data and take them as the input to the decoder. (3) We use the MSE loss function to evaluate the reconstruction residuals.

**Figure 3 sensors-24-06294-f003:**
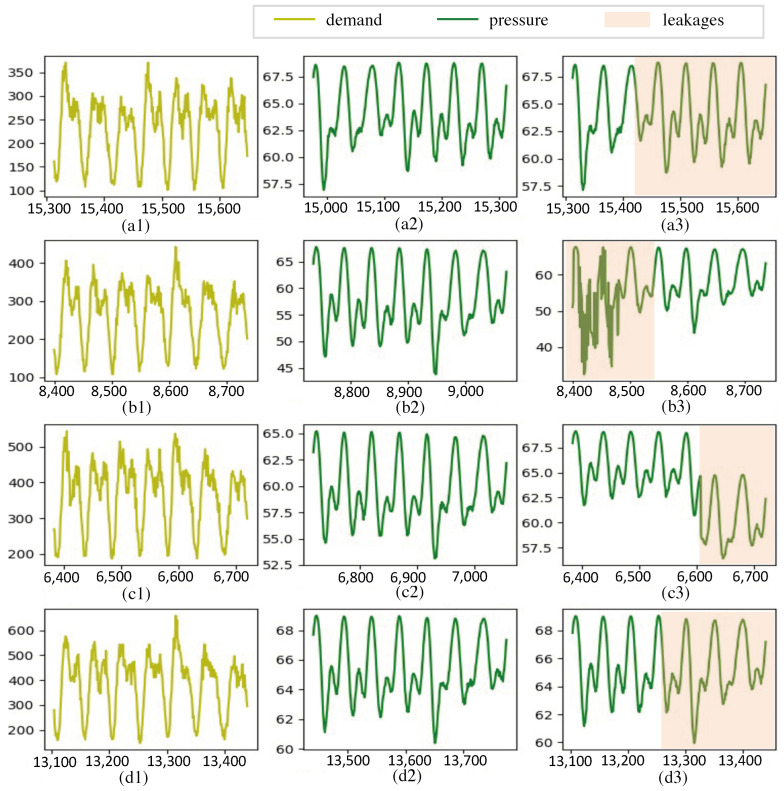
A comparison of pressure data between normal operation and leakage of the pipe network in several scenarios. The left column (**a1**,**b1**,**c1**,**d1**) and middle column (**a2**,**b2**,**c2**,**d2**) show the water demands (in m^3^) and pressure (in m) under normal operation. The right column (**a3**,**b3**,**c3**,**d3**) shows pressure changes in different leakage scenarios.

**Figure 4 sensors-24-06294-f004:**
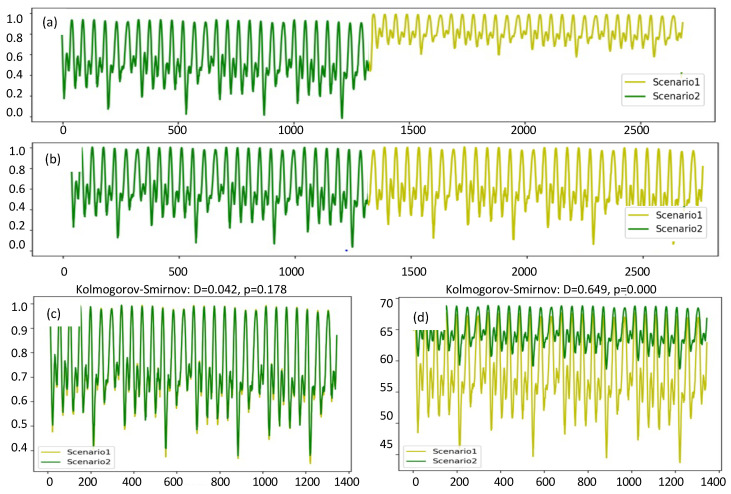
Comparison of pressure (in m, normalized) before and after normalization. Lines with different colors represent different scenarios. (**a**) Concatenated pressure data in different scenarios without normalization. (**b**) Concatenated pressure data in different scenarios with normalization. (**c**) Pressure data with the same distribution according to KS statistics after normalization for each scenario. (**d**) Pressure data with different distributions according to KS statistics.

**Figure 5 sensors-24-06294-f005:**
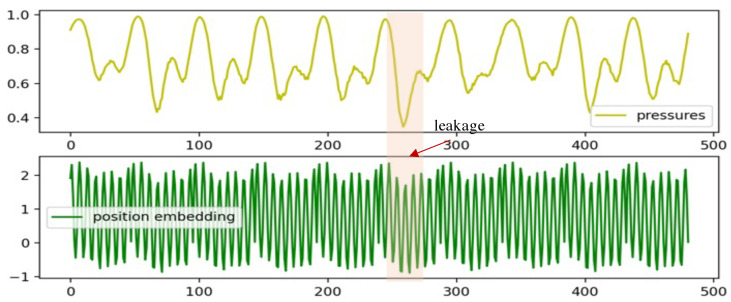
A comparison of pressure data and data with the addition of position embeddings. The top figure illustrates the pressure data (in m, normalized) without the addition of the position embeddings, and the bottom one presents the corresponding pressure data after the addition of the position embeddings. The leakage point is indicated by the area shaded in orange.

**Figure 6 sensors-24-06294-f006:**
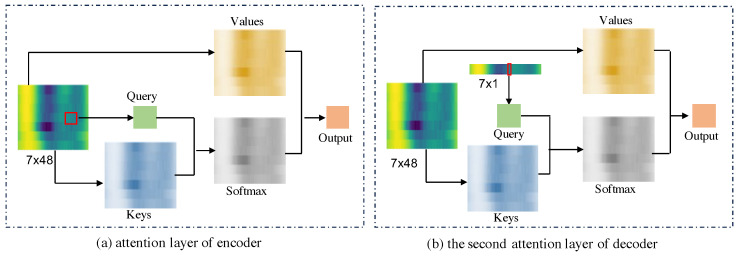
Local attention layers of encoder and decoder. In the attention layer of the encoder, *Q*, *K*, and *V* are obtained from the historical pressure data. In the first attention layer of the decoder, *Q*, *K*, and *V* are obtained from the pressure data at the same time on different days. In the second attention layer of the decoder, *Q* is obtained from pressure data at the same time on different days, and *K* and *V* are obtained from the historical pressure data.

**Figure 7 sensors-24-06294-f007:**
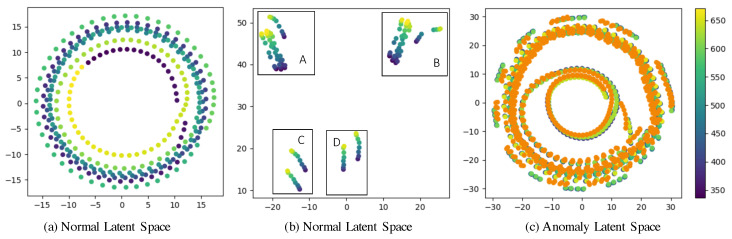
Latent features from normal pressure and leakage data. Each point represents a latent feature at a time step, and the color of the points varies according to the change in the time steps. (**a**) The t-SNE embeddings of the latent features with one week of pressure data; (**b**) the t-SNE embeddings of the latent features at the same hour of different days; (**c**) the comparison of the t-SNE embeddings of the pressure data under normal operation and the data for leakages, denoted by points in green and orange, respectively.

**Figure 8 sensors-24-06294-f008:**
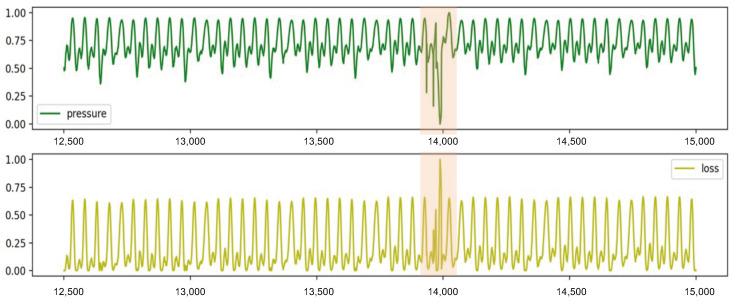
Visualization of leakage detection. The green line denotes pressure data, the yellow line denotes MSE loss between input pressure and reconstructed pressure (in m, normalized), and the orange shade denotes leakages.

**Figure 9 sensors-24-06294-f009:**
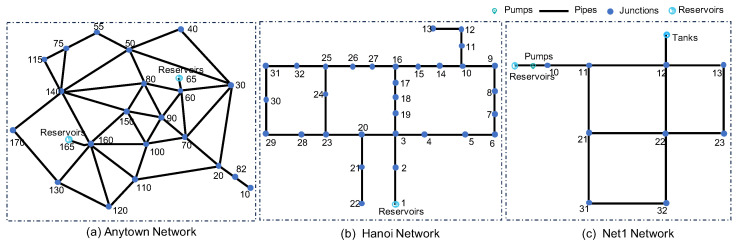
The network topologies of Anytown, Hanoi, and Net1. Each point represents a junction with Pressure Transducer and Flow Transducer.

**Figure 10 sensors-24-06294-f010:**
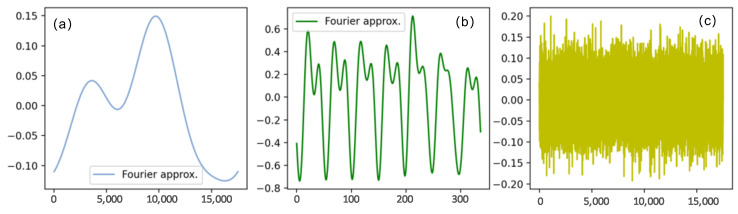
The components of the demand data. (**a**) The fundamental periodicity to simulate the seasonal trends; (**b**) random noise used to generate the variance in daily usage and operations on the pipelines; (**c**) daily consumption-based demand periodicity.

**Figure 11 sensors-24-06294-f011:**
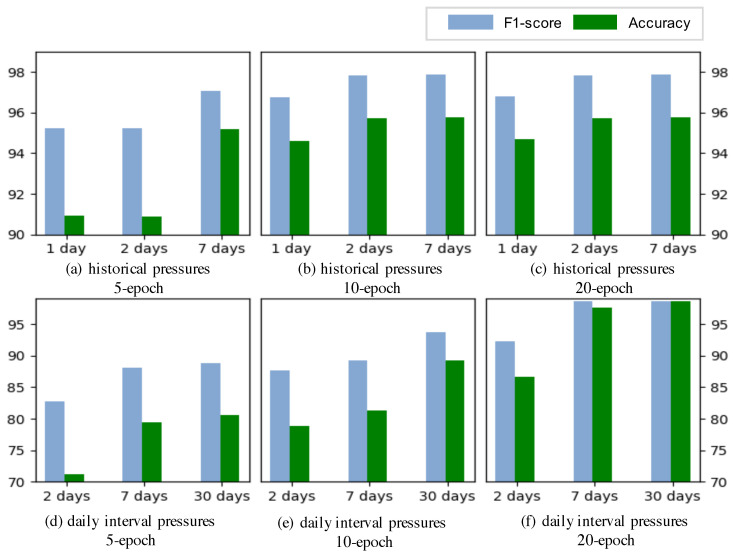
Performance comparison of historical pressure data (**a**–**c**) and pressure data at the same hour on different days (**d**–**f**) for different window sizes. Various training epochs were considered. We use the green color to indicate the performance of “F1-score” and blue to indicate the performance of “Accuracy”. The window size was set to [1 day, 2 days, and 7 days] for the historical pressure data, and to [2 days, 7 days, 30 days] for the pressure data at the same hour on different days.

**Table 1 sensors-24-06294-t001:** Leakage details for the simulation with Hanoi, Net1, and Anytown networks. “S-m” denotes the mth scenario. “A” denotes “Abrupt Leakage”. “I” denotes “Incipient Leakage”. The first row represents various scenarios under different network topologies, and the second row represents the leakage type and leakage duration in the corresponding scenario. There may be different leakages at different times within a scenario in one year.

Scenario	Leakage Details
*Hanoi*	
S-1	A: 4 October 09:00∼9 December 16:00
S-2	I: 15 November 10:00∼14 December 01:30
S-3	I: 21 February 11:30∼11 June 01:00
S-4	I: 18 October 06:30∼2 December 17:00
S-5	A: 18 May 15:00∼15 December 06:30
S-6	I: 8 October 15:00∼31 December 05:00
S-7	A: 7 February 18:30∼9 August 06:00
S-8	A: 23 August 16:30∼21 December 05:30
S-9	A: 31 January 23:00∼29 December 07:30
*Net1*	
S-1	A: 21 April 13:30 08∼17 April 16:30; A: 30 April 11:00∼9 May 13:30
S-2	A: 11 December 21:00∼30 December 07:00
S-3	A: 30 March 20:00∼17 July 09:30; A: 11 December 17:30∼18 December 16:00
S-4	I: 26 July 17:00∼8 September 06:30; I: 10 May 08:00∼8 August 23:00
S-5	A: 11 November 22:30∼3 December 09:30
S-6	A: 16 August 14:30∼18 October 05:30; A: 9 November 04:30∼7 December 22:30
S-7	A: 19 March 07:30∼3 September 12:30; I: 17 June 17:00∼1 September 16:30
S-8	I: 2 October 23:30∼28 November 15:30; I: 29 December 02:30∼30 December 07:30
S-9	A: 12 February 04:00∼31 March 12:30; I: 29 January 04:00∼3 June 03:30
*Anytown*	
S-1	A: 12 December 07:30∼30 December 16:30; A: 28 February 00:00∼7 November 02:30
S-2	I: 25 September 14:30∼27 September 13:30; I: 11 March 00:00∼16 June 04:30
S-3	I: 24 April 19:00∼23 June 11:30; A: 30 June 17:00∼29 October 02:30
S-4	I: 17 May 08:30∼2 December 19:30; I: 11 November 10:00∼14 December 03:30
S-5	I: 17 May 14:30∼28 October 10:00; A: 3 August 03:00∼31 October 02:30
S-6	I: 23 January 18:30∼10 December 09:30; A: 1 February 20:00∼3 October 19:30
S-7	A: 28 November 17:30∼31 December 11:30; I: 10 May 06:30∼7 December 16:00
S-8	I: 24 February 16:00∼29 August 10:30; A: 31 May 23:30∼7 October 02:30
S-9	I: 19 July 06:30∼18 November 19:00; I: 12 December 23:30∼25 December 09:00

**Table 2 sensors-24-06294-t002:** Results of our model on various datasets. “S-m” denotes the mth scenario. “Attention” denotes the simple multi-head self-attention model, which only has the encoder module, and “−1” and “−2” denote the methods where the position embeddings are added to and concatenated with the pressure data.

Dataset		Autoencoder [33]		Attention [40]		Propose-1		Propose-2	
		Acc	F1	Acc	F1	Acc	F1	Acc	F1
Hanoi	S-1	27.94	41.93	90.11	94.80	94.87	97.74	**95.61**	**97.77**
	S-2	79.92	88.44	81.71	89.93	86.16	92.56	**87.58**	**93.38**
	S-3	84.99	89.08	88.50	89.24	**94.29**	**97.06**	94.23	97.02
	S-4	92.20	26.85	**96.12**	**98.02**	92.02	95.84	91.83	95.74
	S-5	98.18	98.87	**98.34**	**99.16**	98.27	99.13	98.20	99.09
	S-6	90.58	77.58	92.72	96.22	96.95	98.45	**97.16**	**98.74**
	S-7	89.91	94.37	90.17	94.83	**95.07**	**97.47**	94.85	97.35
	S-8	93.17	96.32	94.20	97.01	97.46	98.71	**97.85**	**98.91**
	S-9	92.55	96.13	99.09	99.54	**99.46**	**99.73**	99.44	99.72
Net1	S-1	71.88	83.51	79.55	88.61	94.81	97.33	**94.84**	**97.35**
	S-2	72.16	82.86	79.52	88.59	93.09	**95.48**	**93.21**	**95.48**
	S-3	32.31	48.84	82.16	90.21	98.54	99.26	**99.07**	**99.53**
	S-4	31.12	46.05	75.86	86.27	**95.10**	**97.48**	94.86	97.36
	S-5	61.09	75.38	67.46	80.57	90.97	**95.27**	**95.32**	91.49
	S-6	67.39	67.62	69.81	82.22	91.56	78.39	**92.83**	**78.85**
	S-7	83.40	89.72	87.32	93.23	96.93	98.44	**97.06**	**98.51**
	S-8	68.74	25.72	54.89	70.88	**88.22**	**92.80**	88.05	92.70
	S-9	77.92	74.96	89.47	94.44	95.29	97.59	**95.72**	**97.81**
Anytown	S-1	81.07	47.90	83.37	90.93	82.18	90.22	**93.20**	**96.48**
	S-2	27.31	36.44	**81.74**	**89.95**	81.70	89.93	81.70	89.93
	S-3	43.67	60.79	82.38	90.33	92.07	**95.87**	**92.08**	**95.87**
	S-4	64.38	74.21	**97.03**	**98.49**	97.00	98.48	97.00	98.48
	S-5	82.07	84.20	83.42	90.96	90.92	95.24	**90.93**	**95.25**
	S-6	58.51	73.40	95.08	97.47	95.63	97.76	98.29	99.13
	S-7	50.55	67.16	78.38	87.88	89.73	94.59	**95.57**	**97.73**
	S-8	34.21	56.43	82.75	90.56	92.74	96.23	**95.00**	**97.44**
	S-9	48.14	65.00	93.11	96.43	87.13	93.12	**93.08**	**96.41**

**Table 3 sensors-24-06294-t003:** Results of our method compared with the Autoencoder on various datasets. “Attention” denotes the simple multi-head self-attention model that only has the encoder module, and “−1” and “−2” denote the methods where the position embeddings are added to and concatenated with the pressure data. The score is the average of all scenarios shown in Table 2.

Method	Metric	Hanoi	Net1	Anytown
Autoencoder [33]	Accuracy	89.00	62.89	54.43
	F1-score	78.84	66.07	62.83
Attention [40]	Accuracy	93.44	76.22	86.36
	F1-score	96.52	86.12	92.56
Propose-1	Accuracy	94.95	93.83	89.90
	F1-score	97.41	94.67	94.60
Propose-2	Accuracy	95.19	94.55	92.98
	F1-score	97.52	94.34	96.30

## Data Availability

Data are contained within the article.

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
