# Peer review of "A Transformer-Based Approach to Leakage Detection in Water Distribution Networks"

_sensors, 2024, doi:10.3390/s24196294_

Round 1

Reviewer 1 Report

Comments and Suggestions for Authors

Summary:

In the publication, the authors described a method using an artificial neural network model for leakage detection in water distribution network. The model uses a transofmers architecture and consists of an encoder and a decoder. This model can perform classification using only pressure data. The authors used concatenation of positional embedding with input data instead of adding them. The authors conducted tests of the developed solution on 3 network topologies: Hanoi, Net1 and Anytown. The results were compared with the Autoencoder method. The solution using a transformer architecture with concatenation of input data with positional encodings achieved the best F1-score and accuracy metrics in almost every case.

The publication has been prepared with due care, but it is possible to improve it by:

Adding a short section regarding the training of the model, which should include information such as the target number of epochs, whether an early stopping mechanism was used, what optimizer was chosen, etc.

The Conslusion section should be expanded to include the information in Table 3. It should show how the developed solution allowed the Accuracy and F1-score metrics to increase relative to the reference method - Autoencoder.

Author Response

Comments 1: Adding a short section regarding the training of the model, which should include information such as the target number of epochs, whether an early stopping mechanism was used, what optimizer was chosen, etc.

Response 1: Thank you for pointing this out. We agree with this comment. Therefore, we have add a detail regarding the training of the model. Mention exactly where in the revised manuscript this change can be found - the 2nd paragraph in section 5.2.

Comments 2: The Conslusion section should be expanded to include the information in Table 3. It should show how the developed solution allowed the Accuracy and F1-score metrics to increase relative to the reference method - Autoencoder.

Response 2: Thank you for pointing this out. We agree with this comment. Therefore, we have add a detail for this. Mention exactly where in the revised manuscript this change can be found - in section 6.

Reviewer 2 Report

Comments and Suggestions for Authors

This paper proposed a transformer based model for detecting leakages in WDNs. The transformer incorporates an attention mechanism to learn data distributions and account for correlations between historical pressure data and data from the same time on different days, thereby emphasizing long-term dependencies in pressure series. Additionally, we apply pressure data normalization across each leakage scenario and concatenate position embeddings with pressure data in the transformer model to avoid feature misleading. The performance of the proposed method is evaluated using detection accuracy and F1-score.

The study is interesting and valuable. The paper is well organized. Several suggestions and questions are listed as follows:

1) The authors are suggested to give more detailed theoretical basis of the study.

2) More results are suggested to be presented to show the applicability and advantages of the proposed method.

3) The quality of Figures should be improved as the words in some figures are too small.

4) The English language can be further improved.

Author Response

Comments 1: The authors are suggested to give more detailed theoretical basis of the study.

Response 1: Thank you for pointing this out. We agree with this comment. We have revised the Introduction and Related Work sections in the article.

Comments 2: More results are suggested to be presented to show the applicability and advantages of the proposed method.

Response 2: Thank you for pointing this out. We agree with this comment and that is a very good and reasonable question. We have also conducted experimental evaluated on other network topologies, such as the Net3 network. For this paper, we selected three widely used networks in research to present our results.

Comments 3: The quality of Figures should be improved as the words in some figures are too small.

Response 3: Thank you for pointing this out. We agree with this comment. We have revised the figures in the paper.

Comments 4: The English language can be further improved.

Response 4: Thank you for pointing this out. We agree with this comment. Therefore, we have made some changes to the grammar in this paper.

Reviewer 3 Report

Comments and Suggestions for Authors

The research letter entitled " A Transformer based Approach for Leakage Detection in WDNs" is gone through carefully and the comments on the manuscript are appended herewith. In this manuscript, The authors proposed a Transformer-based model for detecting leaks in water distribution networks (WDNs). The model aimed to identify leaks by learning the distribution of normal pressure data, while considering the correlation between historical pressure data and pressure data at the same point in time on different dates. This work sounds very interesting and meaningful, and the analysis is reasonably clear. However, for the paper to be accepted for publication on MDPI, the paper needs to be major revised and the following points need to be addressed:

1What is the basis for setting the time interval for receiving pressure data at half an hour in the paper? It would be helpful to explain the potential effects of using different time intervals in the text.

2The model's real-time processing capabilities, including metrics such as processing speed and response time, should be mentioned in the paper.

3Regarding the impact of window size on leak detection, will the other two networks yield the same results as the Hanoi network?

4Units should be added to parameters such as pressure and time in the figures to enhance readability.

5Some up-to-date articles can be referred to: Nano Energy, 2024, 124: 109498; Nano Energy, 2023, 107: 108132; Energy Conversion and Management, 2022, 269: 116098.

Author Response

Comments 1: What is the basis for setting the time interval for receiving pressure data at half an hour in the paper? It would be helpful to explain the potential effects of using different time intervals in the text.

Response 1: Thank you for pointing this out. In the actual operation of water supply networks, pressure data is typically uploaded to the central station every half hour, and sometimes hourly. In this work, we take these real-world conditions into account. We added some details in Section 5.1.

Comments 2: The model's real-time processing capabilities, including metrics such as processing speed and response time, should be mentioned in the paper.

Response 2: Thank you for pointing this out. We agree with this comment and that is a very good and reasonable question.  Therefore, we have add a detail regarding the training of the model. Mention exactly where in the revised manuscript this change can be found - the 2nd paragraph in section 5.2.  This work primarily focuses on the accuracy of leak detection. In our future work, we will conduct a more detailed analysis of the model.

Comments 3: Regarding the impact of window size on leak detection, will the other two networks yield the same results as the Hanoi network.

Response 3: Thank you for pointing this out. Sure, the model can perform leakage detection using only pressure data, without the need for network partitioning or pipeline parameters.

Comments 4: Units should be added to parameters such as pressure and time in the figures to enhance readability.

Response 4: Thank you for pointing this out. We agree with this comment. Therefore, We have revised the figures in the paper.

Comments 5: Some up-to-date articles can be referred to: Nano Energy, 2024, 124: 109498; Nano Energy, 2023, 107: 108132; Energy Conversion and Management, 2022, 269: 116098.

Response 5: Thank you for pointing this out. We agree with this comment. Therefore, we have revised in this paper.

Round 2

Reviewer 3 Report

Comments and Suggestions for Authors

This paper can be accepted.